# Image-Guided Precision Medicine in the Diagnosis and Treatment of Pheochromocytomas and Paragangliomas

**DOI:** 10.3390/cancers15184666

**Published:** 2023-09-21

**Authors:** Gildas Gabiache, Charline Zadro, Laura Rozenblum, Delphine Vezzosi, Céline Mouly, Matthieu Thoulouzan, Rosine Guimbaud, Philippe Otal, Lawrence Dierickx, Hervé Rousseau, Christopher Trepanier, Laurent Dercle, Fatima-Zohra Mokrane

**Affiliations:** 1Department of Radiology, Rangueil University Hospital, 31400 Toulouse, Francemokrane.fz@chu-toulouse.fr (F.-Z.M.); 2Department of Nuclear Medicine, Sorbonne Université, AP-HP, Hôpital La Pitié-Salpêtrière, 75013 Paris, France; 3Department of Endocrinology, Rangueil University Hospital, 31400 Toulouse, France; 4Department of Surgery, Rangueil University Hospital, 31400 Toulouse, France; 5Department of Oncology, Rangueil University Hospital, 31400 Toulouse, France; 6Department of Nuclear Medicine, IUCT-Oncopole, 31059 Toulouse, France; dierickx.lawrence@iuct-oncopole.fr; 7New York-Presbyterian Hospital/Department of Radiology, Columbia University Irving Medical Center, New York, NY 10032, USA

**Keywords:** pheochromocytoma, paraganglioma, metabolic imaging, radiomic biomarkers

## Abstract

**Simple Summary:**

Pheochromocytomas and paragangliomas are rare neural-crest-derived tumors with variable prognosis. In an era of tailored treatment strategies, medical imaging plays a crucial role in the surgeries, image-guided procedures, chemotherapies, immunotherapies, and radionuclide therapies involved in the diagnosis and treatment of these tumors. Medical imaging helps to confirm the diagnosis, guide surgical resection, assess metastatic staging, and select patients for specific therapies. In this step-by-step review, we perform a comprehensive analysis of recent imaging modalities developed for pheochromocytomas and paragangliomas by covering their content and terminology as well as discussing the future implications of artificial intelligence.

**Abstract:**

In this comprehensive review, we aimed to discuss the current state-of-the-art medical imaging for pheochromocytomas and paragangliomas (PPGLs) diagnosis and treatment. Despite major medical improvements, PPGLs, as with other neuroendocrine tumors (NETs), leave clinicians facing several challenges; their inherent particularities and their diagnosis and treatment pose several challenges for clinicians due to their inherent complexity, and they require management by multidisciplinary teams. The conventional concepts of medical imaging are currently undergoing a paradigm shift, thanks to developments in radiomic and metabolic imaging. However, despite active research, clinical relevance of these new parameters remains unclear, and further multicentric studies are needed in order to validate and increase widespread use and integration in clinical routine. Use of AI in PPGLs may detect changes in tumor phenotype that precede classical medical imaging biomarkers, such as shape, texture, and size. Since PPGLs are rare, slow-growing, and heterogeneous, multicentric collaboration will be necessary to have enough data in order to develop new PPGL biomarkers. In this nonsystematic review, our aim is to present an exhaustive pedagogical tool based on real-world cases, dedicated to physicians dealing with PPGLs, augmented by perspectives of artificial intelligence and big data.

## 1. Introduction

Pheochromocytomas (PCCs) and paragangliomas (PGLs) are both (PPGLs) rare neural-crest-derived tumors. Thanks to the latest advances in medical imaging, diagnoses of PPGLs have increased. Although they share the same histological origin, PCCs and PGLs differ in several aspects, such as functional status, metastatic potential, and risk of recurrence. Improvements in imaging biomarkers and artificial intelligence over the past few decades have offered tailored diagnostic and treatment strategies based on a tumor’s genetic and inherent phenotypic background. This guide provides an innovative image-modality-based approach to PPGLs management through its review of each imaging modality, the specifics of each treatment option, and the current imaging reporting systems available to improve follow-up. Artificial intelligence (AI) has had a major impact on how medical research is conducted. AI requires large, high-quality datasets to create accurate predictive algorithms. This is particularly true for rare conditions such as PPGLs. This review aims to summarize the role of medical imaging in the management of PPGLs, with a special focus on imaging biomarkers and AI.

## 2. Epidemiology, Genetics, and Molecular Background: How to Stratify Risk

### 2.1. Epidemiology and Embryology

PPGLs arise from the adrenal medulla (PCCs) and the extra-adrenal paravertebral sympathetic ganglia (PGLs) of the thorax, abdomen, and pelvis. Head and neck paragangliomas (HaN PGLs) arise specifically from the parasympathetic structures at the base of the skull along the glossopharyngeal (CN IX) and vagal (CN X) nerves. The incidence of diagnosed PPGLs is about 0.2–0.8/100,000 patients/year [1], with a sex ratio close to 1:1, and a reported PCC/PGL ratio of 80%/20% [2]. The prevalence is reported to be 0.2 to 0.6% in adults and 1.7% in children with hypertension. Over the past few decades, the annual age-standardized incidence rates have increased, with a higher age at diagnosis alongside significantly smaller PCC tumor sizes. This is mainly due to improvements in clinical practice and targeted screening plasma and urinary free metanephrine/normetanephrine (M/NM), increased use of imaging studies, and screening of susceptibility-gene-related carriers [3].

### 2.2. An Endocrine Functional Status

PPGLs belong to the neuroendocrine tumors (NETs) group, and produce, store, and secrete one or more catecholamines: M/NM and dopamine. This catecholamine-secreting function is mainly observed in sympathetic-tissue-derived PCCs and PGLs arising below the head and neck (in 96% and 65% of tumors, respectively), whereas the majority (97%) of parasympathetic HaN PGLs are nonsecreting [4]. The catecholamine excess explains the adverse symptoms during initial presentations of NETs (e.g., hypertension, headaches, sweating, and palpitations) as well as their related acute complications (e.g., renal failure, intestinal ischemia, heart attack, angina, pulmonary edema, osteoporosis, and pathological fractures [5,6,7,8]). Due to their endocrine function, plasma and 24 h urinary M/NM measurements are the recommended biochemical tests for PPGLs, both for screening and follow-up, sometimes completed with 3-methoxytyramine serum evaluation. Therefore, screening for catecholamine secretion is critical before any biopsy in case of suspected PPGL, to prevent life-threatening outcomes [9]. However, rare false negatives (small PCCs and nonsecreting PGLs) and medication-induced false positives (diuretics, b-blockers, tricyclic antidepressants, renal failure, acute stress, etc.) limit screening performance [10]. Another plasma protein of interest, chromogranin A (CgA), is stored and released along with catecholamines. Serum CgA levels are elevated in the majority of NETs, including PPGLs, suggesting that serum CgA may be an alternative for screening and follow-up in the outpatient setting, even though medication- and non-medication-induced false positives exist as well. Currently, serum CgA use is limited to preoperative evaluation of NETs patients with otherwise normal M/NM levels and to postoperative follow-up in patients with elevated CgA [11,12].

### 2.3. Genetic Background: A Prerequisite to Understand Pathology

PPGLs are part of inherited tumor syndromes in 40% of cases. This relatively big proportion sets them apart from other NETs and has a major impact on the management of this pathology. More than 20 genes have been identified (such as SDHD, VHL, RET, neurofibromatosis type 1 NF1), including inherited cancer susceptibility genes [4,13,14,15,16,17,18]. Comprehensive molecular research has defined four groups with specific pathways depending on the identified mutated driver and fusion genes: c kinase signaling subtype, pseudohypoxia subtype, Wnt-altered subtype, and cortical admixture subtype [19]. Each genetic underlying mutation has a specific risk of metastatic outcome, multifocal disease, and familial history. Genetic testing is an essential component of not only patient management, follow-up, and prognostication, but also for election of an optimal imaging modality [20]. For instance, metastatic risk is low in PCCs (5–10%), high in PGLs (30–35%), and even higher in the succinate dehydrogenase genetic alterations group (SDHx), such as SDHB mutations carriers (up to 80% in some series) [21]. SDHB and SDHC mutations have autosomal dominant inheritance, at risk of earlier PPGLs (mean age at diagnosis 35–40 years, lifetime risk 30%), with a less differentiated phenotype, a higher risk of metastatic disease at diagnosis or during follow-up [7,22,23,24], and a poorer prognosis [25]. Therefore, early diagnosis of patients and their relatives is critical in improving overall prognosis. Several algorithms based on immunohistochemical staining procedures and genetic screening are currently being implemented in routine practice guidelines [26,27,28].

### 2.4. A Potentially Malignant Tumor with Common Sites of Metastases

Despite the latest progress in the understanding of PPGLs, there is still no histological, molecular, or genetic criteria of malignancy. According to the latest WHO classification [29], malignancy in PPGLs is only defined by proven distant metastases in nonchromaffin tissues (mainly bone, lung, lymph nodes, and liver), even if histologic patterns were reported to be indicative of malignancy, such as small cell or spindle cell patterns [30,31]. In 2017, the American Joint Committee on Cancer (AJCC) proposed a TNM staging [32] classification with a prognostic value (worse survival in stage IV) [33]. According to this classification, the terms “malignant” and “benign” should be abandoned, implying that all PPGLs are classified as having malignant potential [29,34].

PGL metastases are reported to occur earlier than PCC metastases (median age 36 years vs. 46 years), with more synchronous metastatic disease at diagnosis (41% vs. 26%) [35]. Radiologists and nuclear medicine physicians must be aware that the most common metastatic sites are bone (60–70% of metastatic patients), lymph nodes (47%), liver/lung/thorax (38%), and abdomen/pelvis (32%), with PGL metastases mostly spreading to lymph nodes (67%), and PCC metastases mostly spreading to the liver (49–57%) [24,36,37,38].

### 2.5. Prognosis and Prognostic Markers

The 5-year survival rate for cases of malignant PPGLs varies from 50% to 80% [23], with a metastatic risk reported to be 4–5 times higher in PGLs than PCCs [35,39]. In addition to genetic status discussed earlier [25], some factors are reportedly associated with worse survival: male sex, synchronous metastases, larger tumor size (>5 cm), hypersecretion, dopamine secretion phenotype, and serum levels of the dopamine metabolite 3-methoxytyramine [24,30,35,39,40,41,42]. Ki-67 antigen is a nuclear protein expressed only in proliferating cells, and it has a high prognostic value in NETs [43]. This antigen has an unclear prognostic value in PPGLs [30,42,44,45,46], even though it was recently reported to be a significant prognostic factor for locally advanced PCCs [47], and significantly higher in metastatic PCCs [48], with a cut-off of >2–3%.

Several scores are used to better risk stratify patients with PPGLs, based on histological grade (the PASS Pheochromocytoma scaled score, the GAPP Grading of Adrenal Pheochromocytoma and Paraganglioma score [49,50]), or clinical presentation (the ASES score [51]). These scores have high negative predictive values (PASS score up to 99% [31,41,52]), but significantly different survival rates, requiring validation from multicenter clinical trials [49].

The majority of histological criteria are applied to postoperative histological samples. However, preoperative imaging sheds light on poor prognostic factors. For instance, lesions larger than 5 cm are at higher risk of malignancy [40,53]. Additionally, since the GAPP score uses histological cellularity as a main prognostic criterion, MRI diffusion sequences may be a useful alternative for assessing cellularity, although this has yet to be proven [54].

Even if metastatic recurrence risk is higher with locally aggressive tumors, it is also present in patients with sporadic variants (5-year risk: 1%). Those patients should also receive lifelong follow-up because of long-term tumor recurrence risk (20-year risk: 6.5%), even if this risk is lower compared to patients with hereditary tumors (20-year risk: 38%) [55].

## 3. Anatomical Imaging Techniques in Initial Diagnosis

### 3.1. How to Explore an Indeterminate Adrenal Mass?

Although PCCs may be discovered when working up patients with a catecholamine syndrome and abdominal pain, they can also be incidental [56]. For example, 4% of incidentally discovered adrenal masses are proven to be a pheochromocytoma [57]. For adrenal incidentalomas, the European Society of Endocrinology recommends performing a clinical exam, biological screening, and conventional imaging (noncontrast CT for adults, or abdominal MRI otherwise). Benign features on noncontrast CT imaging include attenuation value <10 Hounsfield units (HU), size <4 cm, and homogeneity. If all criteria are met, no further imaging is required. Otherwise, a contrast-enhanced CT examination or MRI with chemical shift imaging is necessary [58,59].

### 3.2. Additional Value of Contrast-Enhanced Computed Tomography Scan 

A contrast-enhanced computed tomography scan (CE-CT scan) assesses washout characteristics of adrenal masses after enhancement by comparing the attenuation values at specific times and calculating percentages of absolute and relative washouts (Figure 1). An absolute enhancement washout ≥60% and relative enhancement washout ≥40% are 96–100% specific for adenoma. False positives are rare but have been reported, such as lipid-rich PCC [60] or renal metastases with increased washout [61].

### 3.3. Additional Value of Magnetic Resonance Imaging with Chemical Shift Imaging

This magnetic resonance imaging (MRI) technique is an alternative to CE-CT scan, and takes advantage of adenomas’ high lipid content, which induces variations in signal intensity (SI) out-phase (op) compared to the in-phase (ip). These values are compared to the spleen’s to calculate the adrenal lesion-to-spleen ratio (ASR) and adrenal signal intensity index (ASII). ASR is either calculated in publications as (SIop adrenal mass/SIop spleen)/(SIip adrenal mass/SIip spleen), to assess remaining signal expected to be inferior to 0.71 for the diagnosis of adenoma [62], or as [(SIop adrenal mass/SIop spleen)/(SIip adrenal mass/SIip spleen) − 1], to calculate percentage of signal loss, expected to be inferior to −35.9% [63] (Figure 2). ASII is an alternative calculated as (SIip adrenal mass − SIop adrenal mass)/(SIip adrenal mass) × 100 (signal remaining). Diagnosis of adenoma is achieved when SII is superior to 16.5% (1.5 T) [64].

### 3.4. Pheochromocytomas and Paragangliomas: Variable Morphological Characteristics Using Anatomical Imaging

CT and MR examinations provide important information on preoperative localization of a tumor, even if their characterization performance may be limited. PPGLs are reported to have avid enhancement after contrast agent injection and a slow washout. Around half of PPGLs are reported to be homogeneous and hypointense in T1-weighted (T1-w) and markedly hyperintense in T2-weighted (T2-w) MRI sequences compared to spleen and liver, with a possible “salt and pepper” appearance related to flow voids in tumor vessels. The rest of the PPGLs are more heterogeneous with mixed areas of high and low signal intensity, which are possibly cystic or swirl-like areas that correspond to hemorrhage or necrosis on histopathology (Figure 3). However, enhancement patterns and aggressive features of other solid tumors (i.e., ill-defined contours, heterogeneity, etc.) did not have any relationship with potential risk of malignancy [62], even if these were predicted by radiomics (discussed later).

### 3.5. Head and Neck Parangangliomas: Specific Concerns 

HaN PGLs can occur in several common locations: carotid body 60% (angle of contact with carotid vessels), vagal nerve (osseous involvement of skull base), jugular bulb, and tympanic (extension to the middle ear) [65]. 

Multiparametric MRI and MR angiography is the best initial imaging modality in HaN soft tissue tumors, with sensitivities and specificities in HaN PGLs, respectively, of 90–95% and 92–99% [66]. MRI protocol should include axial T1-w sequence without fat suppression, axial T2-w fast spin-echo, and 3D T1-w contrast-enhanced with fat suppression sequence. By providing a larger anatomic coverage (from aortic arch to skull base) compared to conventional MRI sequences, contrast-enhanced MR angiography (CE-MRA) and dynamic contrast-enhanced (DCE) should also be used if there is clinical evidence of PGL (pulsatile mass, screening in first degree relatives): HaN PGLs demonstrate early initial avid enhancement (arterial tumor blush) distinctive from other cervical benign lesions, and shorter time to peak followed by washout pattern (type-III curve) [67,68]. Diffusion-weighted imaging (DWI) is also useful in distinguishing HaN PGLs from other benign tumors, with a relatively lower apparent diffusion coefficient (ADC) (1.17 to 1.25 × 10^−3^ mm^2^/s) [67,69]. Even if DCE (based on capillary perfusion) and ADC (based on cellularity) can be considered MRI biomarkers, no differences in signal were observed between sporadic and SDHx-related HaN PGLs [69].

CT scan and CT angiography can also be performed for diagnosis and to delineate anatomic relations, but compared to MRI, CT is mainly useful in assessing the degree of bone destruction, especially in the skull base.

Conventional arteriography, historically used for diagnosis and localizing HaN PGLs, is now reserved for embolization in specific cases [70].

### 3.6. Specific Concern: How to Manage Asymptomatic SDHx Mutation Carriers? 

Early detection of PPGLs in first-degree relatives with inherited genetic mutation is crucial. During childhood, clinical (blood pressure, symptoms) and biological assessment (M/NM plasma and 24 h urinary measurements) are performed for initial screening along with MRI to reduce cumulative radiation exposure (HaN, thorax, abdomen, and pelvis) rather than molecular imaging computed tomography (which should only be performed in adults) [71]. In cases of negative initial screening, annual clinical follow-up is recommended, with biologic follow-up every two years and MRI screening every 2–3 years [71].

## 4. Molecular Imaging Techniques in Precise Diagnosis and Follow-Up

Several molecular imaging techniques are used to diagnose and follow-up PPGLS (Appendix A) with specific diagnostic, prognostic, and theranostic advantages. 

### 4.1. Metaiodobenzylguanidine: About the Historical Tracer 

Metaiodobenzylguanidine (MIBG) is a noradrenalin analogue with similar metabolic uptake to catecholamines. Its uptake is mediated by norepinephrine transporters, followed by vesicular storage using monoamine transporters VMAT type 1 and 2. MIBG is either labelled with ^123^I or 131I for metabolic imaging. MIBG scintigraphy has long been the gold standard for molecular imaging of PPGLs (Figure 4). However, it has several limitations compared to other types of functional imaging, such as poor spatial resolution and sensitivity for small lesions due to the detection limits of conventional gamma camera imaging, reduced specificity due to physiologic uptake in normal adrenal glands, risk of medication interference, and low uptake in malignant PPGLs. MIBG scintigraphy performance is reportedly good in PCCs and abdominal PPGLs (Se 74–97%) [72], but tends to be poor in thoracic and HaN PGLs, which are mostly V-MAT1 negative [73,74,75]. However, the most important indication and use of ^123^I-MIBG with single photon emission computed tomography (SPECT/CT) remains assessing eligibility for 131I-MIBG therapy. If positron emission tomography with computed tomography (PET/CT) is not available, then MIBG is the next best exam for detection of sporadic PCC. MIBG can also be used to diagnose inherited PCC or multifocal and/or metastatic PGL or to assess PPGL relapse after a surgery, when anatomical imaging (CT/MRI) is limited (scars, anatomical distortion, artefacts of metallic clips) or by preference with a positive baseline exam.

### 4.2. Contributions of Computed Tomography Using Dopamine and Glucose Analogues 

Dopamine D2 receptors are membrane norepinephrine transporters predominantly expressed in sympathetic structures, which have strong avidity for monoamines [76]. Pretreatment with carbidopa has been shown to increase tracer uptake and increase the sensitivity of PPGLs detection [77] as well as diminishing the physiological background uptake, allowing for a better tumor/background ratio. Two types of dopamine analogues are routinely used for PPGLs. ^18^F-Fluorodopamine (FDA) is an analogue of dopamine that uses norepinephrine transporter-mediated cellular uptake. It has excellent performance for detecting and localizing PCCs in patients with known disease (Se 98%, Spe 100%) [73], metastatic PPGLs (MPPGLs) [73,78,79,80], and HaN PGLs, including SDHx-related tumors [81]. FDA is of little help in cases of nonsecreting PCCs [82].

^18^F-dihydroxyphenylalanine (^18^F-DOPA) is structurally similar to the dopamine precursor L-DOPA and enters neuroendocrine cells through a large amino acid transporter. Its lack of uptake in normal adrenal glands (contrary to MIBG) allows for good differentiation between a normal adrenal gland and PCC based on their respective standardized uptake value (SUV) [83]. With sensitivity and specificity greater than 80%, ^18^F-DOPA is reportedly more efficient than MIBG scintigraphy [72,73,84] in detecting PPGLs, especially if combined with CT/MRI [85,86]. This increased efficiency can be used to better detect biological catecholamine excess, even in patients taking medications that confound M/NM testing [87]. Currently, ^18^F-DOPA use is recommended for nonmetastatic sporadic and inherited PCCs except SDHx-related (NF1, RET, VHL, and MAX) [88].

In PPGLs, increased glucose uptake is not an exclusive marker of tumoral dedifferentiation, as it can also be attributed to a specific genetic defect and/or hypoxic-induced phenotype (HIF) [89]. For example, in VHL patients (including those with nonmetastatic disease), the HIF pathway can induce a pseudohypoxic metabolic shift that increases anaerobic glycolysis, called the Warburg effect [90]. This is responsible for the high glucose uptake seen in ^18^F-FDG PET, which does not correlate with tumor differentiation. The use of FDG for inherited PPGLs is reportedly able to differentiate between cluster 1 and cluster 2 PPGLs, with higher uptake in cluster 1 PPGLs [91]. Therefore, ^18^F-FDG PET has poor sensitivity for detecting nonmetastatic sporadic PCCs (58%), but appears to be as sensitive as ^18^F-DOPA and better than MIBG scintigraphy for detecting MPPGLs (82%) [89,92,93,94] (Figure 5). Its sensitivity is also reportedly increased in SDHx- and VHL-related PPGLs (Se 92–99%) [93,95], and better than ^18^F-DOPA in metastatic SDHB PPGLs [95]. Overall, FDG with PET/CT may be used for most PPGLs, except perhaps for HaN PGLs; however, if ^18^F-DOPA or ^68^Ga-SSTa are available, then these tracers would be a better choice for most of the indications [88].

### 4.3. Positron Emission Tomography with Computed Tomography Using Somatostatin Analogues

Somatostatin analogues (SSTas) are small regulatory peptides with a high affinity for cellular somatostatin receptors (SSRs), which carry out hormonal functions such as inhibition of growth hormone secretion in the pituitary and gastropancreatic systems, inhibition of tumor growth, and apoptosis activation. NETs usually have a high density of SSRs, explaining their avidity for SSTa-derived agents. SSR subtypes commonly expressed in PPGLs are SSR-1 (90%), SSR-2 (70%), and SSR-3 (80%) [96]. SSTas are medically used in several ways: bound to a radiotracer for functional imaging or to a radionuclide for peptide receptor radionuclide therapy (PRRT). A potential therapeutic hormonal option, already used in gastroenteropancreatic (GEP) NETs, is also being assessed in the LAMPARA study (NCT03946527) for MPPGLs [97]. Octreotide and Pentetreotide are commonly used in scintigraphy with single photon emitters (Octreoscan) and reportedly have increased affinity for SSR-2 compared to MIBG, especially for HaN PGLs (Se97%, Spe82%) [98,99,100,101,102,103].

Radiometal positron emitting tracers, mainly represented by ^68^Gallium (^68^Ga) labelled with SSTas, are used for PET-CT imaging. These tracers allow for accurate initial pretherapeutic staging, early detection of relapse, and treatment candidate assessment for patients with nonresectable NETs (Figure 6). In SDHB-related and sporadic PPGLs, ^68^Ga-DOTATATE PET/CT provides a 98.6% lesion-based detection rate, significantly higher than all other imaging modalities [104,105,106]. 

Therefore, ^68^Ga-DOTATATE imaging has become the radiotracer of choice in MPPGLs, HaN PGLs, and SDHx-related PGLs [88] as it is more accurate and less irradiating for the patient. However, it currently does require a specialized radiopharmacy unit and an expensive generator, which limits its widespread availability.

### 4.4. Current Guidelines for Molecular Imaging in Diagnosis and Staging of PPGLs

Initial molecular imaging aims to confirm the diagnosis, guide potential surgical resection, assess metastatic staging, and select patients for PRRT by choosing an appropriate modality based on the clinical context. The European Association of Nuclear Medicine Practice Guidelines [88] were updated in 2019 with an additional focus dedicated to molecular imaging and theranostics in NETs in 2021, reaching consensus for the use of ^68^Ga-DOTA-SSTa PET/CT in staging and restaging of suspected extra-adrenal PPGLs [107]. 

### 4.5. Future Perspectives in Metabolic Imaging

With regard to the future of SSTa imaging, ^18^F-SiFAlinTATE is a new PET radiotracer developed for NET imaging that appears to be a better alternative to ^68^Ga-labelled SSTa because of its lower production cost and longer half-life [108]. However, further studies are needed to better understand its exact role in PPGLs detection and follow-up.

Despite a lack of internalization, SSTr subtype 2 antagonists (called BASS and JR11) were reported to provide higher tumor uptake and better tumor-to-liver background ratios than agonists. Preliminary results show improved imaging of ^68^Ga-NODAGA-JR11 PET/CT compared to ^68^Ga-DOTATOC PET/CT in GEP NETs. Using ^177^Lutetium (^177^Lu) as radiotracer, higher tumor doses on PRRT can be obtained with ^177^Lu-DOTA-JR11 compared to ^177^Lu-DOTATE in metastasized NETs [109].

## 5. Planning a Surgical Treatment: The Role of Imaging

Surgery is intended to be curative for nonmetastatic tumors. Even though there are no curative treatments for MPPGLs, therapeutic options also exist for cases of symptomatic and/or progressive disease, such as palliative surgery, systemic therapeutics, image-guided treatments, and external or isotope-mediated radiotherapy [110]. As we will discuss, data are scarce and large clinical trials are lacking specifically for PPGLs, because of the extremely low prevalence of metastatic cases. Instead, some therapeutics have been tested on heterogeneous NET cohorts, including those originating from different organs, despite inherent limitations.

### 5.1. Reminder on Biopsy and PPGLs

Even if radiologists should not biopsy PPGLs, this may occur in cases of alternative expected diagnosis. We reiterate that 24 h urinary M/NM measurements are critical before any biopsy of a lesion in an anatomical location suspicious for PPGL, to prevent life-threatening outcomes [9]. 

### 5.2. Curative Surgical Management: How to Prepare a Surgery?

Accurate preoperative staging, performed with both anatomical and molecular imaging, is crucial to identify anatomical risks, assess extent of disease, and discuss surgical options [44] (Figure 7).

For example, with HaN PGLs, Shamblin surgical classification [111] of carotid body tumors (CBT) is based on extension to carotid vessels. Preoperative imaging criteria based on CE-CT scan or MRI, such as tumor volume, angle of contact with arterial structures, presence or absence of fat plane interface between tumor and arterial adventitia, presence of peritumoral veins, and distance from skull base, were reported to correlate to Shamblin group [112,113]. These findings and the detection of anatomical variants help in surgical planning, predicting resection difficulties, and operative outcomes. 

In abdominal PPGLs, the laparoscopic approach is usually first because of its many advantages: minimally invasive, decreased blood loss and morbidity, and faster recovery. However, open surgery with laparotomy is still required for large tumors or tumors closely in contact with major blood vessels (interaortocaval PGL or retrocaval tumor extension) [114] (Figure 8). After complete resection, the risk of recurrence is estimated to be 5% over 5 years of follow-up (new tumor 22%, local recurrence 23%, metastatic recurrence 55%) [115]. To improve the early detection of all types of recurrence, annual follow-up is recommended after surgery of nonmetastatic PPGL for at least 10 years after complete resection, or lifelong if there are other high-risk factors (young age, large tumor, PGL, genetic disease) [116,117].

### 5.3. Palliative Surgery: The Debulking Strategy Concept

A cytoreductive surgical approach can be used for MPPGLs, to improve symptoms of catecholamine excess or resect a mass in a critical anatomical location [118,119]. Palliative surgery in MPPGLs was reported to significantly improve median OS (148 months versus 36 months), even in patients with nonsecreting tumor [119]. Furthermore, surgery could enhance the concentration of radionuclides in remaining metastatic sites, reducing tumoral burden prior to PRRT.

## 6. Imaging Guided Therapeutic Options

Percutaneous ablation of adrenal tumors, PPGLs, and metastases (mainly in bones and liver) is a minimally invasive treatment option with short-term efficacy [120,121]. There is increasing interest around ablative therapies mostly as a palliative treatment due to their ability to reduce tumor burden and catecholamine excess in MPPGLs [122,123]. For instance, the early diagnosis and treatment of bone metastases is a therapeutic challenge, as more than 72% of affected patients will suffer from skeletal-related events (pathological fracture, severe pain, spinal cord compression) responsible for the loss of independence or a poor quality of life [38]. Several studies report improvements in metastases-related symptoms, pain, and prevention of skeletal-related events (SRE) with percutaneous cementoplasty, osteosynthesis, or thermal ablation [124].

### 6.1. Thermal Ablation Techniques for Percutaneous Tumor Destruction

Among various thermoablation techniques currently used [125,126,127], radiofrequency ablation (FRA) is a widespread technique [128,129] based on frictional heating, and results in coagulative necrosis [130]. This procedure is frequently used on local oligometastatic liver ablations in NETs [131], including MPPGLs, where it reportedly improved hypertensive symptoms and metastasis-related pain [123,132]. However, based on multidisciplinary discussion, this technique is reserved for oligometastatic lesions. The expected pattern of imaging changes on follow-up is a larger lesion on CT scan at 3 to 6 months (safety margin of ablated tissue), followed by a shrinkage of the remaining RFA-treated area with a possible remaining scar [131]. Some authors consider necrosis achieved if the lesion has no significant enhancement (<10 UH) on 6-month follow-up CT [132].

### 6.2. Transarterial Chemoembolization for Liver Metastases 

Several studies have expressed interest in using transarterial chemoembolization (TACE) on the hepatic lesions of NETs to improve both OS and symptoms [133]. In MPPGLs, TACE is expected to benefit hypertension, tumor size, and plasma M/NM levels [134,135,136,137]. The usual protocol involves use of *mitomycin C* or *epirubicin*, with tumor shrinkage expected in the following months [134].

### 6.3. Transarterial Embolization with Polyvinyl or Ethylene Vinyl Alcohol (Onyx) 

In HaN PGLs, polyvinyl or Onyx embolization are preoperative or palliative options that reduce blood flow to jugulo-tympanic and vagal PGLs [138], with tumor volume stability reported to be achievable in 75% of cases at long-term follow-up [139]. For carotid body PGLs, Onyx embolization reduces blood loss and operative time, although it can make surgical dissection more difficult [140].

### 6.4. Percutaneous Ethanol Injection 

This imaging-guided technique was reported as an alternative to surgery in benign PCC by inducing necrosis and reducing tumor size [141]. It is also of interest in treating metastases [122]. However, there is a clear need for comparative studies and further data to assess its benefit on OS.

### 6.5. External Radiotherapy: Local Control and Symptoms Improvement 

For HaN PGLs (temporal bone, carotid body, and/or glomus vagal) and intrathoracic PGLs, radiotherapy (RT) is an alternative to surgery, especially in cases of extensive spread when tumor resection would present neural and vascular risk. Surgical resection of jugular and vagal PGLs generates significantly more cranial nerves palsies and major complications with less tumor control compared to radiotherapy, suggesting that surgery should be considered only for selected cases [142,143]. In MPPGLs, external beam radiation therapy (EBRT) is also useful in obtaining local control and improving symptoms [144,145]. Anatomical imaging (especially head and neck MRI) and molecular imaging are crucial to adapt the radiation field to limit radiation-induced complications (xerostomia, neural deficits, osteonecrosis), and for post-treatment follow-up. Expected findings on imaging are growth control (mainly progression control, or reduction in size) and decrease in vascularity rather than tumor elimination [144].

## 7. Systemic Therapies: Impact of New Therapeutics in Imaging Management

Several concepts support the use of systemic therapies for PPGLS. These therapies are intended to have an antiprogression effect (as chemotherapy), a symptomatic effect (by managing catecholamine excess with metyrosine and other antihypertensive medications), and pain reduction. Several types of systemic therapies exist for PPGLs [32,117,146].

### 7.1. Cytotoxic Therapies 

The classic cytotoxic chemotherapies cyclophosphamide, vincristine, and dacarbazine, known as the CVD-protocol, can be used to treat PPGLS. They tend to decrease catecholamine excess and reduce tumor and lymph node size in 30 to 70% of patients [147,148,149,150]. The clinical benefit to overall survival in MPPGLs is unclear, even if research has reported longer mean survival in responders (3.8 years) compared to nonresponders (1.8 years) [148]. Alternatives to the CVD-protocol are now first-line treatment options, such as temozolomide, which provides a significantly longer progression-free survival (PFS) in SDHB-mutated MPPGLs compared to other MPPGLs [151]. As with other NETs, assessing treatment response to cytotoxic drugs using anatomic imaging with RECIST is limited by the slow-growing behavior of PPGLs. Therefore, a decrease in size is rarely achieved in NETs, even for patients with the best survival rates [151,152,153,154]. Radiologists should be aware that contrast medium injection protocols could interfere with tumor size measurements [152], thus biasing image interpretation [155]. Since assessing a metabolic response is important and could be better detected with metabolic imaging, it is likely this imaging modality could be more suited to therapy monitoring [156].

### 7.2. Targeted Therapies 

Several targeted therapies, such as tyrosine kinase inhibitors (TKI) and interferon alpha biotherapy [157], are also in use. For instance, sunitinib (TKI group) has become a first-line option in MPPGLs, inducing decreased tumor size and improvement in hypertension [158,159], and has recently provided promising results in PFS at 12 months (ongoing clinical trial FIRSTMAPP) [160]. Interferon alpha has also shown improved disease stabilization [157]. Even if SSTa has a significant effect on hormone levels in GEP NETs [161,162] with decreased symptoms of flushing and diarrhea, their use in PPGLs is currently not being investigated. As most targeted agents are cytostatic, there is a need for image-based criteria to assess tumor response. For example, perfusion CT’s ability to describe the change in tumor density of hypervascular NETs is of interest [154,163]. This could also apply to PPGLs since they are hypervascular slow-growing tumors; however, further comparative studies are needed.

### 7.3. Immune Checkpoint Inhibitors 

The use of immune checkpoint inhibitors (ICIs) for NETS is currently under investigation in clinical trials. For MPPGLs, pembrolizumab was recently reported to induce a 43% nonprogression rate and a 75% clinical benefit rate (CBR) at 27 weeks [164]. Due to their distinct effect on antitumor immunity, there are many well-documented immune-related phenomena associated with ICIs use. Several new patterns of tumor response and progression (pseudoprogression, hyperprogression, abscopal effect), as well as adverse events are widely described in the literature and are important for radiologists to know, because they can lead to misdiagnoses [165,166,167,168,169,170,171]. In fact, detection of immune-related adverse events (iRAE) in patients treated with ICIs is a crucial challenge for radiologists [172]. They can occur at any site during patients’ treatment with ICIs, and mainly affect the endocrine glands (hypophysitis, thyroiditis, hepatitis, pancreatitis), lung and mediastinal lymph nodes, digestive tract (enterocolitis), and joints (arthralgia). 

iRECIST introduces the concepts of immune unconfirmed progressive disease (iUPD), which states that an increase in the size of a lesion or the appearance of new lesions should be closely followed-up on by subsequent imaging. If on the next radiological examination there is no change in the size or appearance of the tumor, then it remains as iUPD. However, if the size increases, then it is classified as immune confirmed progressive disease (iCPD). A decrease in size is classified as conclude response (iCR, iPR). Also developed for ICIs, the iPERCIST (immune PET response criteria in solid tumors) criteria are a modified classification system that takes into consideration a novel type of tumor response based on a dual time point, called unconfirmed progressive metabolic disease (UPMD), which helps to limit false interpretations. If confirmed at a repeat scan 3–4 weeks later, the tumor is classified as confirmed progressive metabolic disease (CPMD). In non-small-cell lung cancer, one-third of UPMDs were later classified as SMD or PMR (stable or partial metabolic response), thereby using iPERCIST to provide prognostic information [173].

## 8. Targeted Radionuclide Therapies in Palliative Treatments

### 8.1. Rationale 

These molecular-imaging-based modalities take advantage of the fact that peptide-linked radiotracers can target membrane-bound receptors. The two main purposes of these include (1) confirming sufficient radiotracer affinity to the tumor, and (2) binding to the specific membrane receptor, which allows internalization into the tumor and can emit radiation that leads to DNA damage and apoptosis. Targeted radionuclide therapies (TRT) have a low antigenicity, rapid tissue penetration, fast blood clearance, and a relatively easy and less expensive method of synthesis compared to monoclonal antibodies. They have become an effective alternative to chemotherapies, with fewer side effects. Their use is mostly in patients with inoperable or metastatic NETs.

### 8.2. Iobenguane

Iobenguane or 131I-MIBG is reported to improve symptoms from catecholamine excess in over 40% of MPPGLs patients and induce a mean progression-free survival of 23.1–28.5 months for common and mild adverse events [174,175,176]. The expected radiologic pattern of response is mainly stable disease (SD) or partial response (PR) on CT scan [174], and a lower MIBG uptake (Figure 9). Therapy with higher delivered doses of 131I-MIBG has been reported to have higher 5-year OS, but should be reserved for selected patients because of increased adverse events (pulmonary toxicity, myelosuppression, myelodysplasia, leukemia) [177]. Currently this tracer’s variable regional, national, and international availability poses a problem for its widespread access. Recently, the FDA has approved the novel high-specificity-activity (HDSA)-131I-MIBG therapy, which looks very promising but is not yet available in most countries. More randomized studies could establish its place in therapeutic management [175].

### 8.3. Peptide Receptor Radiotherapy 

The key concept behind this therapeutic approach is that a radionuclide can bind a peptide that specifically targets a cellular receptor [178,179,180]. Specific data in patients with PPGLs are scarce, but an ongoing clinical trial is assessing peptide receptor radiotherapy (PRRT) in inoperable PPGL, with results expected in 2023 (NCT03206060). ^90^Yttrium in ^90^Y-DOTATOC is also reportedly effective on response rate (morphological response in 10–40% of patients [181]), survival time, and symptomatic response in NETs, with limited adverse events [182,183]. ^177^Lu-DOTATATE, considered a third-generation SSR-PRRT, has the advantage of increased affinity for octreotate compared to octreotide on SSR-positive tissues, allowing longer tumor residence [184] for a higher tumor-absorbed dose and promising results in PFS for GEP-NETs [185]. As compared to ^90^Y, the main advantage of ^177^Lu is that it is not a pure beta-emitter, it also emits low-energy gamma rays, allowing for post-therapy SPECT imaging and dosimetry. The NETTER-1 trial is also worth mentioning here, even though it was performed in midgut NETs, because it is a randomized phase III clinical study offering high-level evidence of efficacy with ^177^Lu-DOTATATE on PFS and response rate [186]. In HaN PGLs, ^177^Lu-DOTATATE was also reported to be an adequate alternative for achieving PR or SD in cases where surgery or radiation were contraindicated due to local neurovascular structures [187]. With regard to NETs, a specific threshold of intensity of uptake with ^68^Ga-SSTa has not been validated. A practical approach is mostly used, applying the four-point scale of Rotterdam initially developed for the octreoscan with at least an uptake equal to or higher than physiological liver uptake for PRRT eligibility. Given the choice between both tracers, it will very likely depend on a pragmatic approach based on the imaging results of MIBG and the SSA radiotracer, with priority given to the agent with the highest uptake. Equivocal factors can be considered, like the toxicity profile, the risk factors, and, of course, the more practical aspects related to availability, reimbursement, insurance, and experience [188].

## 9. Tumor Response Management: New Concepts and Pitfalls 

MPPGLs are slow-growing and heterogeneous, which has important implications for diagnosis and management. 

First, baseline imaging is crucial in assessing disease extent, planning treatment, and guiding further management in order to reduce long-term treatment side effects. Second, RECIST 1.1 does not accurately assess early tumor response, and a more precise and reliable set of guidelines is needed [189]. One potential alternative, computer science, uses semiautomated measurements such as advanced segmentation, and can provide quantitative analysis by recognizing and delineating a lesion, which provides an estimate of tumor volume or can calculate tumor burden. Third, performing several imaging modalities (by combining anatomical and molecular imaging) on a heterogeneous tumor can be useful for follow-up. Fourth, if a new, highly sensitive imaging modality (i.e., new PET tracers) that was not used at baseline shows new lesions, they should be considered "new baseline lesions" for this imaging modality instead of disease progression. Finally, considering that these tumors are slow-growing and most of their treatments are “noncytoreductive”, disease nadir and baseline tumor burden should be incorporated into management.

### 9.1. RECIST 1.1 Limitations

These quantitative size-based radiological criteria are used for therapy response assessment with standardized choice and target lesions quantification [190]. Several situations can lead to nonreproducible measures, including irregular lesions with complex shapes, bone lesions, fibrosis or necrotic lesions with unprecise limits, variability of behavior of lesions for a same patient, arbitrary selection of target lesion leading to imprecise therapeutic response evaluation, and slow-growing tumor. Additionally, some new classes of treatment provide mechanisms and patterns of response that are not size-assessable. For example, ICIs can induce an effective but delayed response, or behave like a progressive lesion because of an initial immune-mediated flare in size. Antiangiogenic agents can also induce hemorrhage, necrosis, and myxoid degeneration, with a possible early transitory increase in size [191].

### 9.2. Molecular Imaging Reporting and Data System: The SSTR-RADS

SSTR-targeted radiotracers allow specific imaging of SSTR-avid structures with some limitations (normal organs uptake, urinary excretion, inflammatory diseases, etc.) that should be known by radiologists [179]. SSTR-RADS is the first PET classification system based on tumor uptake, and it uses a reliable five-point scale that can assess treatment response and identify ideal candidates for PRRT from poor responders [192,193]. This is a promising system that, however, needs validation in large prospective studies, as other response criteria, such as positron emission tomography response criteria in solid tumors (PERCIST) and European organization for research and treatment of cancer (EORTC) criteria, are currently only validated for response assessment with ^18^F-FDG PET/CT. 

## 10. Current and Future Perspectives in the Era of Artificial Intelligence 

As the aim of this review is to propose a pedagogical content, we truly believe that an extensive knowledge of genetic background, screening strategies, diagnostic tools, and treatment options has to be treated in this review, even if not recent. Moreover, many centers do not have access to radiotracers, and still have to manage those patients with alternative imaging techniques. For these reasons, we explain in this review that a key concept for artificial intelligence is to collect good-quality data, i.e., complete, highly informative imaging. This point is, from our point of view, crucial, needing a perfect knowledge of these tumors and their complete management. As with other NETs, PPGLs are highly heterogeneous neoplasms, with many variations at the genetic, cellular, molecular, functional, clinical, and histopathological levels.

In order to create more tailored treatments, these different variables need to be understood as a global interrelated system. The fact that there is such genetic and phenotypic heterogeneity suggests that there is potential to develop personalized treatments. For example, predictive biomarkers could be used to identify patients most likely to respond to treatment, assess the effectiveness of PRRT in MPPGLs for protocol adjustment, or help choose alternative/combined treatments with optimal cost/benefit [194,195].

### 10.1. Imaging, Radiomics, and Biomarkers as Predictors of Tumor Type and Progression

Recently, several dedicated mathematical models have been developed to improve the characterization of undetermined adrenal mass, for example, to differentiate adrenocortical adenoma from carcinoma [196]. An other predictive calculation model based on imaging characteristics on unenhanced CT, such as sharp-edged necrosis, unsharp necrosis, ring sign, and spherical shape, also had good specificity and sensitivity (80%/95%) for diagnosing PCCs [197]. Machine-learning-based quantitative texture analysis (QTA) could also differentiate subclinical PCCs from lipid-poor adenomas [198,199] or be a useful tool to early identify malignant PPCs subtypes from CE-CT scans [200]. Several anatomical biomarkers have also been proposed to improve tumor monitoring, and others are still being validated (Appendix A) [201]. 

Volume assessments for early detection of change also provide better interoperator agreement compared to measuring the long tumor diameter according to RECIST, which leads to better detection of early partial responders versus progressive disease [202]. Tumor growth rate (TGR) in GEP-NETs was reportedly an early predictor of PFS [203,204] because it revealed a large proportion of patients with active tumor growth despite classification as SD according to RECIST 1.0. This was especially true during early treatment response evaluation.

SPECT and PET metabolic biomarkers mainly assess tumor burden based on volumetric lesion segmentation, and also provide several parameters of prognostic value such as SUV mean, SUV max, metabolic tumor volume (MTV), and total metabolic tumor volume (TMTV). These are easily extracted biomarkers obtained using traditional visualization software, reportedly associated with overall survival (OS), PFS, and response prediction [205].

At the voxel level, textural features (TFs) such as entropy, homogeneity, and intensity variation have been proven to correlate with tumor aggressiveness. They are also of interest for risk stratification in NETs. In GEP-NET patients undergoing pretreatment, SSTR-PET CT, entropy, and intensity may predict OS [206], and repeated SSTa PET/CT may also be used for therapy monitoring for early prediction of response [207].

### 10.2. Metabolomics 

Metabolomics, or metabolite profiling, is the comprehensive analysis of small-molecule metabolites to assess phenotypic and genetic characteristics of a lesion. It links molecular genetics concepts to imaging. Several new markers are described in the literature. Magnetic resonance spectroscopy (MRS) for in vivo metabolomics is a highly promising field. For instance, 1H-MRS SUCCES (SUCCinate Estimation by Spectroscopy) detects succinate concentrations to noninvasively measure mutations in the biomarker SDHx. Succinate concentration can be increased up to 100 times compared to non-SDHx-mutated PGLs, which allows for detection of in vivo SDH deficiency in most PPGL patients. This technique has limitations for those with a small tumor, hemorrhagic or necrotic spots, or respiratory motion on imaging, and because there is no widespread availability or experience for this technique. This modality could allow for early detection of SDH deficiency in routine clinical practice, quicker than genetic tests, with a positive impact on management and clinical outcomes. It could prompt a search for tumor SDHx mutations in cases of negative germline genotyping with a positive succinate peak and detect them before surgery or confirm a suspicious lesion as metastatic. Furthermore, in the setting of a suspected SDHx mutation, a known predisposition to other tumors such as GIST could optimize CT/MRI imaging analysis [208,209,210,211].

### 10.3. Genomic and Methylomic 

At the genome scale, DNA methylation is a major method of regulating gene transcription and phenotype expression. Hypermethylation in promoted gene regions can induce gene silencing, with possible pro-oncogenic shift of regulation pathways. Recent findings indicate that methylome remodeling caused by SDHx mutations results in major transcriptional abnormalities with significant impact on epigenetics features. This makes methylome and transcriptome new biomarkers of interest directly linked to phenotypic features such as metastatic or recurrence and various survival rates depending on methylation clusters [212]. In NETs, for example, the NETest is a 51-multigene assay performed by four different prediction algorithms based on PCR analysis of a peripheral blood sample of specific NET circulating transcripts. This reflects real-time tumor activity and is reportedly of predictive value since it was associated with response type to SSA-PRRT [213] or diagnosing progression of disease one year earlier than image-based evidence [214]. Preliminary studies have shown a high accuracy of this test for PPGLs. However, this test is expensive, the exact mechanism is not known, it is not reimbursed, and it needs to be validated in prospective randomized studies [215,216]. In PPGLs, specifically, integrated bioinformatics analysis based on microarrays identified in PCCs has also found disease-causing genes of diagnostic, prognostic, and therapeutic value (good prognosis for KCNH2, KCNQ2, KCNQ1; poor prognosis for SCN2A) [217,218].

MiRNAs physiologically regulate gene expression by targeting mRNAs to induce their degradation and/or repression. In PPGLs, miRNome datasets identified alterations and miRNA signatures of prognostic value. Six miRNAs (miR-21-3p, miR-183-5p, miR-182-5p, miR-96-5p, miR-551b-3p, and miR-202-5p) were predictive of shorter time to progression and were associated with metastatic risk in PPGLs [219].

## 11. Conclusions

Pheochromocytomas and paragangliomas are rare neural-crest-derived tumors with variable prognosis. This comprehensive analysis of recent imaging modalities and terminology is crucial to understand the current step-by-step approach from diagnosis to treatment of these tumors (Figure A1). The main aim of this nonsystematic pedagogical review of the literature is to familiarize physicians with all the aspects of these tumors. A tumor’s genetic and inherent phenotypic background has a major impact on several aspects of the disease, such as functional status, metastatic potential, and risk of recurrence. Tailored diagnostic and treatment strategies including surgery, image-guided procedures, chemotherapies, immunotherapies, and radionuclide therapies require specific medical imaging modality-based approaches (Figure A2). Metabolic imaging management and modalities, with specific imaging reporting systems, have improved follow-up and selection of therapeutic options. 

However, several challenges remain, such as clinical relevance of imaging biomarkers and artificial intelligence value to identify new parameters. Further multicentric studies are needed, as well as data aggregation, to improve overall patient management. To this end, considering all these aspects is crucial in order to achieve high-quality patient management and good-quality data to apply artificial intelligence at a large scale.

## Figures and Tables

**Figure 1 cancers-15-04666-f001:**
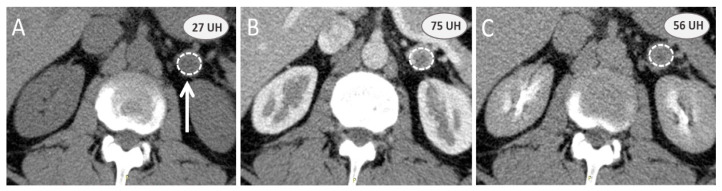
Adrenal contrast-enhanced computed tomography scan washout calculation: a noninvasive option to assess adrenal mass. Computed tomography examination is performed on a 54-year-old male patient to evaluate an incidental adrenal mass (white arrow) with a nonenhanced attenuation value (**A**) greater than 10 UH (27 UH). Absolute and relative washouts based on arterial (**B**) contrast-enhanced and 10 min (**C**) delayed phases are, respectively, 50% and 32%, and do not favor benign adrenal adenoma. The final diagnosis after surgery was a sporadic nonsecreting pheochromocytoma.

**Figure 2 cancers-15-04666-f002:**
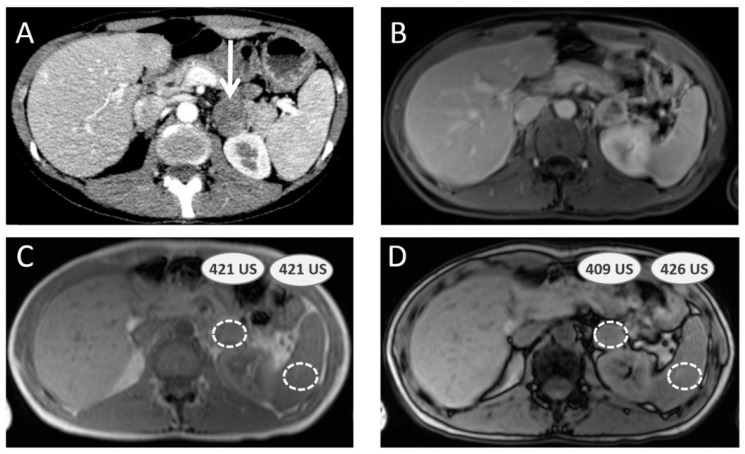
Adrenal-to-spleen magnetic resonance signal ratio calculation to assess adrenal mass. A 40-year-old female patient with multiple endocrine neoplasia type 2A syndrome presents with a left adrenal mass (white arrow) on computed tomography scan (**A**). Magnetic resonance imaging is performed, including gadolinium-enhanced sequence (**B**) and chemical shift sequences ((**C**) IN unit signal values in adrenal mass and spleen parenchyma; (**D**) OUT unit signal values in same spots), providing an adrenal-to-spleen ratio with remaining signal of 0.96, which is over the benign adenoma cut-off (0.71). Surgery confirmed pheochromocytoma.

**Figure 3 cancers-15-04666-f003:**
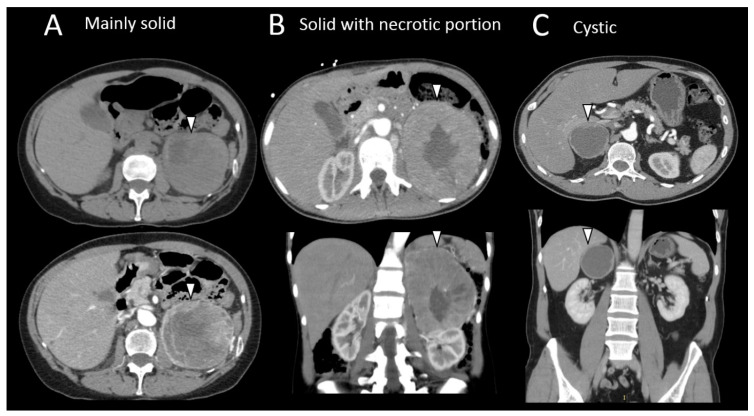
Different radiological morphologic presentation of pheochromocytomas. (**A**) Computed tomography imaging of a large mainly solid left pheochromocytoma (arrow head) in a 63-year-old female patient suffering adrenergic syndrome (takotsubo syndrome, drug-resistant hypertension). (**B**) Axial, coronal, and sagittal computed tomography scan slides of a high-volume left pheochromocytoma with necrotic center (arrowhead) in a 32-year-old female patient with multiple endocrine neoplasia type 2 syndrome. (**C**) Axial and coronal computed tomography scan slides of a large, cystic right pheochromocytoma (arrowhead) in a 47-year-old male patient.

**Figure 4 cancers-15-04666-f004:**
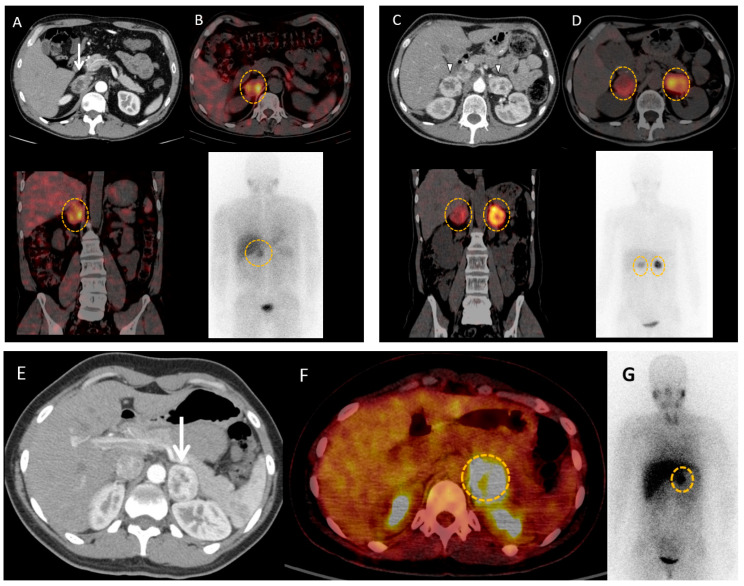
MIBG imaging in pheochromocytomas and paragangliomas. (**A**,**B**) 46-year-old male patient with neurofibromatosis type 1 syndrome and hypertension resistant to medications with a right adrenal mass (white arrow) on computed tomography scan (**A**). ^123^I-MIBG scintigraphy (**B**) shows important uptake in the adrenal mass (orange circles). Adrenalectomy confirmed pheochromocytoma, and induced resolution of hypertension. (**C**,**D**) 49-year-old female patient with Von Hippel Lindau syndrome is discovered to have bilateral adrenal masses (white arrow heads) on computed tomography scan (**C**) with important uptake on ^123^I-MIBG scintigraphy (orange circles) (**D**). Surgery confirmed bilateral pheochromocytomas. (**E**–**G**) 39-year-old female patient with history of cervical paraganglioma, and biological catecholamine excess, is discovered to have a left adrenal mass (white arrow head) on computed tomography (**E**), showing significant uptake on ^18^F-FDG positron emission tomography with computed tomography (**F**) and ^123^I-MIBG scintigraphy (orange circles) (**G**). Surgery provides final diagnosis of pheochromocytoma.

**Figure 5 cancers-15-04666-f005:**
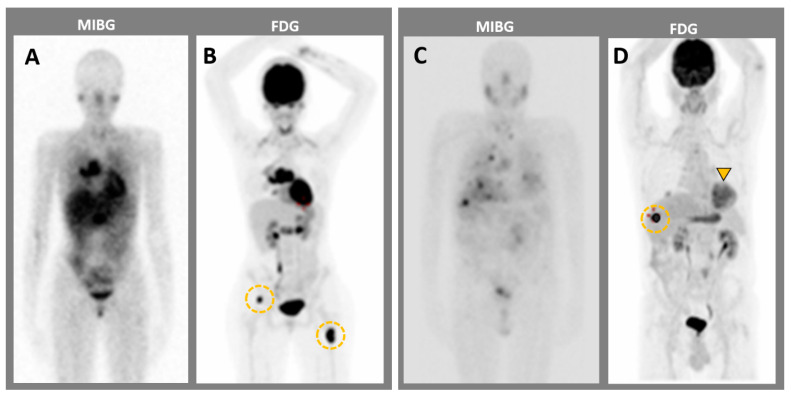
MIBG imaging compared to ^18^F-FDG PET imaging in two patients with metastatic pheochromocytomas. (**A**,**B**) Patient 1. MIBG (**A**) has a lower sensitivity for bone metastases than ^18^F-FDG PET (orange circles). (**B**–**D**) Patient 2. MIBG imaging shows (**C**) less sensitivity for both adrenal mass (orange arrowhead) and liver metastasis (orange circle) as compared to ^18^F-FDG PET (**D**).

**Figure 6 cancers-15-04666-f006:**
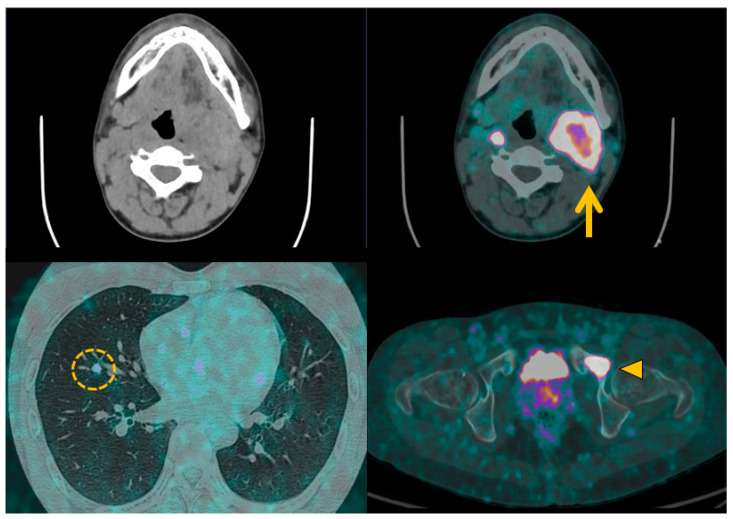
^68^Ga-DOTATOC positron emission tomography with computed tomography in malignant pheochromocytomas and paragangliomas. Patient with metastatic left cervical paraganglioma (orange arrow) received ^68^Ga-DOTATOC positron emission tomography imaging prior to ^177^Lu-DOTATATE peptide receptor radiotherapy. Metabolic imaging helps to identify several metastases in lung (orange circle) and bone (orange arrowhead) to validate the indication for therapy, and provides a pretherapeutic baseline imaging for later comparison.

**Figure 7 cancers-15-04666-f007:**
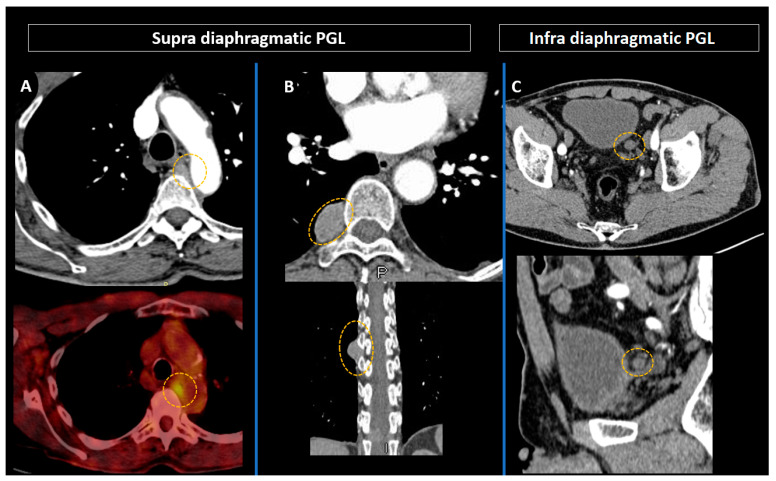
Preoperative combined anatomical and molecular imaging in paragangliomas (orange circles). (**A**) Intrathoracic paraganglioma in an 86-year-old patient with multiple endocrine neoplasia type 2A syndrome. (**B**,**C**) Latero-esophageal (**B**) and latero-vesical (**C**) paragangliomas in a 66-year-old patient with neurofibromatosis type 1 syndrome.

**Figure 8 cancers-15-04666-f008:**
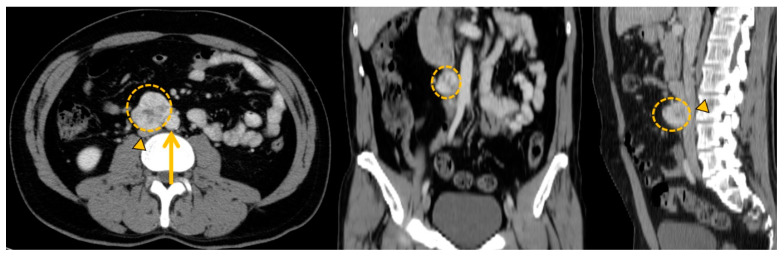
Anatomical imaging to prepare curative surgical project with vascular risks. Zuckerkandl organ paraganglioma (orange circle) in a 44-year-old female patient presenting direct contact with major blood vessels (aorta: orange arrow and inferior cava vein: orange arrowhead).

**Figure 9 cancers-15-04666-f009:**
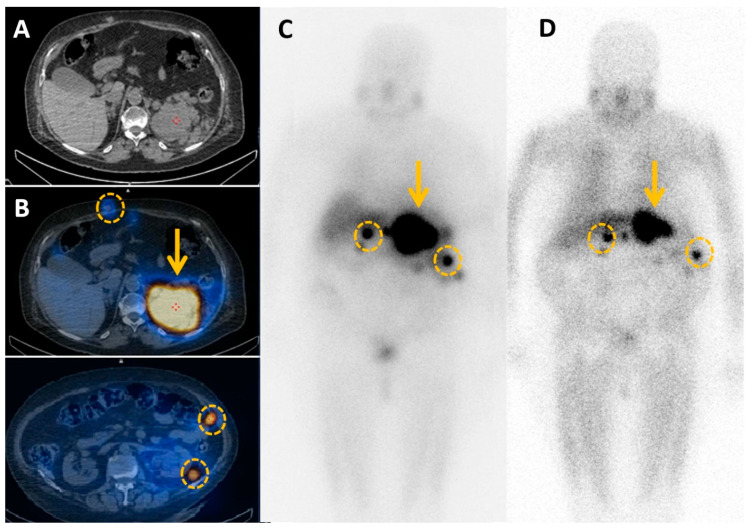
Metabolic imaging for local and systemic extension. Patient treated for a large left pheochromocytoma by surgery is diagnosed nine years later with a local relapse on computed tomography scan (**A**). MIBG imaging confirms local relapse (**B**) (orange arrow), but also subcutaneous and peritoneal metastases (orange circles). Debulking strategy would be too aggressive, so 131I-MIBG therapy is proposed. Post-therapy MIBG scintigraphy (**D**) shows a reduced size of the mass and peritoneal implants, and lower MIBG uptake compared to pretherapeutic MIBG scintigraphy (**C**).

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
