# Peer review of "Image-Guided Precision Medicine in the Diagnosis and Treatment of Pheochromocytomas and Paragangliomas"

_cancers, 2023, doi:10.3390/cancers15184666_

Round 1
Reviewer 1 Report
The present review addresses the recent imaging modalities developed for pheochromocytomas and paragangliomas. The manuscript is well written, and comprehensive. However, there are some major issues that the authors need to amend before the manuscript can be considered for publication.
Title.
In my opinion the title is too long (25 words) and it needs to be shortened in order to be more attractive to the readers.
Abstract.
I really liked the simple summary; however, I found the abstract somehow confusing. The authors presented the challenges and current imaging directions in the field but they did not present clearly the topics of the review. From the abstract the reader should get a broad overview on what is following in the manuscript. The term artificial intelligence should be annotated in the abstract (it is the main purpose and the new knowledge in the field!!!) and further presented. Please reorganize the abstract section so it is clearer and more comprehensive.
Also, spell out the abbreviations used in the abstract (eg. PPGL, NET, AI), otherwise it’s really difficult to understand.
Introduction.
The introduction section is unsatisfactory. It should catch the attention of the reader and show the importance of the review. I would expand the introduction section by focusing more on why it is timely to write a review on that topic and what the author should expect by reading it.
Main text.
This review presents and summarizes a great amount of information about imaging techniques (some of them are already known for many years) and it does not focus on the latest discoveries in the field. The review should focus more on introducing the imagine strategies in the era of artificial intelligence, which is the most interesting and new information that will be attractive to the readers. I would try to condense some already known information and expand the recent advances more.
The flow between the sections has to be improved. The review is separated in many sections and subsections and the paragraphs are not correlated with each other. The review should not be just description of the topic and the terms. Currently, the manuscript reads more as a state of facts and techniques rather than a continuous topic.
References.
A great number of references are old and outdated. It is not acceptable to use references more than 5 years ago especially concerning the epidemiology and description and the recent monitoring techniques of the disease. Please update literature with primary research articles from the last 5 years.
Author Response
Reviewer #1:
-R1-1: “In my opinion the title is too long (25 words) and it needs to be shortened in order to be more attractive to the readers. “
Answer :We thank the editor for this very important comment. An alternative title could be “Image-Guided Precision Medicine in the Diagnosis and Treatment of Pheochromocytomas and Paragangliomas” to simplify it.
-R1-2: “I would expand the introduction section by focusing more on why it is timely to write a review on that topic and what the author should expect by reading it“.
Answer : We thank the editor for this comment. The introduction has been modified accordingly.
Page 4: « Medical field has been marked in recent decades by the advent of artificial intelligence (AI) which has revolutionized several medical concepts, particularly in imaging. In-deed, AI changes the way of designing medical imaging, which must become as homogeneous as possible in order to collect high quality data to be able to use it for AI purposes. This is all the more true for rare conditions, as PPGLs. This review aims to propose a first-approach synthesis on those topics for practitioners, medical students involved in the care of these patients. A special focus on imaging biomarkers and artificial intelligence will also be discussed in this comprehensive review.”
-R1-3: “The review should focus more on introducing the imaging strategies in the era of artificial intelligence, which is the most interesting and new information that will be attractive to the readers. I would try to condense some already known information and expand the recent advances more. “
Answer : as the aim of this review is to propose a pedagogical content, we truly believe that an extensive knowledge of genetic background, screening strategies, diagnostic tools and treatment options -have to be treated in this review, even if not recent. Moreover, many centers do not have access to last radiotracers, and still have to manage those patients with alternative imaging techniques. For these reasons, we explain in this review that a key concept for Artificial intelligence is to collect good quality data ie complete, highly informative imaging. This point is, in our point of view crucial, needing a perfect knowledge of these tumors and their complete management. This point has been emphasized both in introduction section (page 4) and the comment above has been added to artificial intelligence chapter. (page 21, 10. Current and future perspectives in the era of artificial intelligence)
-R1-4: “The flow between the sections has to be improved. The review is separated in many sections and subsections and the paragraphs are not correlated with each other. The review should not be just description of the topic and the terms. Currently, the manuscript reads more as a state of facts and techniques rather than a continuous topic. “
Answer : We thank the reviewer for this comment. The entire flow of the article has been rebuilt, and several sections have been renamed in order to improve the flow.
-R1-5: “A great number of references are old and outdated. It is not acceptable to use references more than 5 years ago especially concerning the epidemiology and description and the recent monitoring techniques of the disease. Please update literature with primary research articles from the last 5 years.
Answer : We thank the reviewer for this comment. All redundant references dating from more that 5 years were removed. However, some references are still up to date reguarding the important breakthrough in PPGLs knowledge. For instance, article Gimenez-Roqueplo and al, with PMID: 14500403 “ Mutations in the SDHB Gene Are Associated with Extra-Adrenal and/or Malignant Phaeochromocytomas », published in 2003 introdcued a key concenpt in PPGLs management and is still up to date. Finally, all recent references concerning the topic were cited in this manuscript.
Reviewer 2 Report
The manuscript sounds technically average; however, I have following concerns should be addressed before any decision.
1. Please explain in your captions of figure and title of table, why are these tables or figures necessary in your paper? What are the purposes and what are the message you want to deliver via these figures and tables?
2. The current metrics might not be sufficient to judge the performance of the model holistically. Please enhance the result analysis part of your paper.
3. The existing literature should be classified and systematically reviewed, instead of being independently introduced one-by-one.
4. In the introduction section, the motivations of the proposed access control model must be included in detail. The section numbering must be changed in the paper organization paragraph.
5. The abstract is too general and not prepared objectively. It should briefly highlight the paper's novelty as what is the main problem, how has it been resolved and where the novelty lies?
6. The 'conclusions' are a key component of the paper. It should complement the 'abstract' and normally used by experts to value the paper's engineering content. In general, it should sum up the most important outcomes of the paper. It should simply provide critical facts and figures achieved in this paper for supporting the claims.
7. For better readability, the authors may expand the abbreviations at every first occurrence.
8. The author should provide only relevant information related to this paper and reserve more space for the proposed framework.
9. The theoretical perceptive of all the models used for comparison must be included in the literature.
10. What are the real-life use cases of the proposed model? The authors can add a theoretical discussion on the real-life usage of the proposed model.
11. The related works section is very short and no benefits from it. I suggest increasing the number of studies and add a new discussion there to show the advantage.
12. The descriptions given in this proposed scheme are not sufficient that this manuscript only adopted a variety of existing methods to complete the experiment where there are no strong hypothesis and methodical theoretical arguments. Therefore, the reviewer considers that this paper needs more works.
13. Key contribution and novelty has not been detailed in manuscript. Please include it in the introduction section
The paper needs minor correction.
Author Response
Reviewer #2:
-R2-1: “Please explain in your captions of figure and title of table, why are these tables or figures necessary in your paper? What are the purposes and what are the message you want to deliver via these figures and tables? “
Answer: We thank the reviewer for this comment. While figures are good examples of real-life patients’ management, we agree with the reviewer that tables are too extensive. For this reason, all tables were edited and added as supplementary material. (se answer R3-4).
-R2-2: “The current metrics might not be sufficient to judge the performance of the model holistically. Please enhance the result analysis part of your paper. “
Answer: We thank the reviewer for this comment. Nevertheless, this study does not contain a model that can be judged in results analysis. Indeed, as a non-systematic review aiming a pedagogical approach for a better comprehension of PPGLs management in clinical routine, this manuscript does not allow any quantitative results.
-R2-3: “The existing literature should be classified and systematically reviewed, instead of beingindependently introduced one-by-one.
Answer: thank the reviewer for this comment. In this study, we have chosen to conduct a nonsystematic review of literature, in a pedagogical approach. Each reference corresponds to the topic of the paragraph where it is cited. We believe that this type of approach can reach a high number of readers, especially when newly dealing with PPGLs.
-R2-4: “In the introduction section, the motivations of the proposed access control model must be included in detail. The section numbering must be changed in the paper organization paragraph.
Answer: the introduction has been modified (see answer R1-2)
-R2-5: “The abstract is too general and not prepared objectively. It should briefly highlight the paper's novelty as what is the main problem, how has it been resolved and where the novelty lies? “
Answer: we thank the reviewer for this comment. The abstract has been remodeled and modified.
-R2-6: “The 'conclusions' are a key component of the paper. It should complement the 'abstract' and normally used by experts to value the paper's engineering content. In general, it should sum up the most important outcomes of the paper. It should simply provide critical facts and figures achieved in this paper for supporting the claims. “
Answer: We thank the reviewer for this comment. Indeed, the conclusion section has been modified accordingly and important “sum up” figures were added to this section.
-R2-7: “For better readability, the authors may expand the abbreviations at every first occurrence. “
Answer : We thank the reviewer for this comment. Abbreviations were detailed at first occurrence in main text, but not in abstract and introduction. We added and abbreviation list at the beginning of the article of the main text as suggested.
-R2-8: “What are the real-life use cases of the proposed model? The authors can add a theoretical discussion on the real-life usage of the proposed model. “
Answer: We thank the reviewer for this comment. In this study, we have chosen to conduct a nonsystematic review of literature, in a pedagogical approach. Real-life cases are shown as examples using detailed figures of real patients, treated in our institution.
-R2-9: “The related works section is very short and no benefits from it. I suggest increasing the number of studies and add a new discussion there to show the advantage. “
Answer: We thank the reviewer for this comment. In this study, we have chosen to conduct a nonsystematic review of literature, in a pedagogical approach.
-R2-10: “The descriptions given in this proposed scheme are not sufficient that this manuscript only adopted a variety of existing methods to complete the experiment where there are no strong hypothesis and methodical theoretical arguments. Therefore, the reviewer considers that this paper needs more works “
Answer: We thank the reviewer for this comment. Nevertheless, this study does not contain a model that can be judged using metrics Indeed, as a non-systematic review aiming a pedagogical approach for a better comprehension of PPGLs management in clinical routine, this manuscript does not allow any quantitative results. As a pedagogical review, our manuscript has to present a wide range of existing methods, and do not use any theoretical argument.
Reviewer 3 Report
The present review analyzes the role of imaging in the diagnosis and treatment of Pheochromocytomas and Paragangliomas (PPGL), through a comprehensive revision of each imaging modality, the specifics of each treatment option and the current imaging reporting systems available to improve follow-up. Moreover, a special focus on imaging biomarkers and artificial intelligence is also discussed.
It is a comprehensive and analytic review, that summarizes the most used imaging modalities, their clinical and therapeutical application, and the possible future use of machine learning. However, the manuscript should be extensively revised, mainly to improve the readability of the paper.
1) In the "Simple summary" and "abstract" sections, the abbreviations are used without explaining the meaning. Please add the meaning of the abbreviations.
2) In the simple summary, at the beginning of the paragraph, it is written metastatic pheochromocytoma and paraganglioma; it is better to remove "metastatic" since the focus of the review is on pheochromocytoma in general and not only on metastatic PPGL.
3) Please read and check extensively the text. There are abbreviations without explaining the meaning, abbreviations used in the title of the paragraph (see paragraph 3.2, 3.3), references at the end of the title paragraph (see 3.1), abbreviations used in the figure legend (please write the entire word, for example in figure 2 NEM2A is known as MEN2A…).
4) The tables are unreadable and should be modify especially in the layout. In table 1 there is a problem with the alignment (check the SSTa section); moreover, the content of the table should be more homogenous, the abbreviations should be explained in the footnotes. The same for Table 2, it is unreadable.
5) Check the Figures and figure legend (Figure 6 is unreadable, it is suggested not to use the abbreviations in the figure legend)
6) I suggest adding an abbreviations list at the beginning of the main text.
7) Paragraph 10: remove "discussion" at the beginning of the title paragraph
8) The text should be extensively and carefully revised (grammar mistakes, sentences without the verb).
9) Figures A1 and A2 are not mentioned in the text, please check.
10) Please cite these 3 studies related to radiomics in adrenal masses and long-term follow-up in PPGL:
1- Crimì, F.; Agostini, E.; Toniolo, A.; Torresan, F.; Iacobone, M.; Tizianel, I.; Scaroni, C.; Quaia, E.; Campi, C.; Ceccato, F. CT Texture Analysis of Adrenal Pheochromocytomas: A Pilot Study. Curr. Oncol. 2023, 30, 2169-2177. https://doi.org/10.3390/curroncol30020167.
2- Torresan F, Crimì F, Ceccato F, Zavan F, Barbot M, Lacognata C, Motta R, Armellin C, Scaroni C, Quaia E, Campi C and Iacobone M. “Radiomics: a new tool to differentiate adrenocortical adenoma from carcinoma”. BJS Open. 2021;5(1):zraa061. doi: 10.1093/bjsopen/zraa061.
3- Torresan, F.; Beber, A.; Schiavone, D.; Zovato, S.; Galuppini, F.; Crimì, F.; Ceccato, F.; Iacobone, M. Long-Term Outcomes after Surgery for Pheochromocytoma and Sympathetic Paraganglioma. Cancers 2023, 15, 2890. https://doi.org/10.3390/cancers15112890
The text should be extensively and carefully revised (grammar mistakes, sentences without the verb).
Author Response
Reviewer #3:
-R3-1: “In the "Simple summary" and "abstract" sections, the abbreviations are used without explaining the meaning. Please add the meaning of the abbreviations. / I suggest adding an abbreviations list at the beginning of the main text “
Answer: We thank the reviewer for this feedback, abbreviations were initially detailed at first occurrence in main text, but not in abstract and introduction. We added and abbreviation list at the beginning of the article of the main text as suggested.
-R3-2: “In the simple summary, at the beginning of the paragraph, it is written metastatic pheochromocytoma and paraganglioma; it is better to remove "metastatic" since the focus of the review is on pheochromocytoma in general and not only on metastatic PPGL. “
Answer : thank you for this feedback, text was corrected according to this comment.
-R3-3: “There are abbreviations without explaining the meaning, abbreviations used in the title of the paragraph (see paragraph 3.2, 3.3), references at the end of the title paragraph (see 3.1) : °abbreviations used in the figure legend (please write the entire word, for example in figure 2 NEM2A is known as MEN2A…).
Answer: We thank the reviewer for this comment. All unnecessary abbreviations were edited (see answer R2-7).
-R3-4: “The tables are unreadable and should be modify especially in the layout. In table 1 there is a problem with the alignment (check the SSTa section); moreover, the content of the table should be more homogenous, the abbreviations should be explained in the footnotes. The same for Table 2, it is unreadable.
Answer: We thank the reviewer for this comment. Indeed, tables are too detailed for the content of the manuscript. Thereby, they were removed from the core of the manuscript and added as supplementary material.
-R3-5: “Check the Figures and figure legend (Figure 6 is unreadable, it is suggested not to use the abbreviations in the figure legend)
Answer: We thank the reviewer for this comment. All figures were edited. All figure legends were checked. Figure 6 has been modified as well as its legend. Unnecessary abbreviations were edited.
-R3-6: “The text should be extensively and carefully revised (grammar mistakes, sentences withoutthe verb)
Answer: We thank the reviewer for this comment. All the manuscript has been extensively checked by a native English speaker in medical field.
-R3-7: “Figures A1 and A2 are not mentioned in the text, please check.
Answer: We thank the reviewer for this comment. Figures A1 and A2 are cited only conclusion. Indeed, these two figures were built as a visual summary of the manuscript, allowing a global overview of PPPGLs imaging management using a multidisciplinary approach
-R3-8: “Please cite these 3 studies related to radiomics in adrenal masses and long-term follow-up in PPGL […]:
Answer: We thank the reviewer for this comment. Indeed, radiomics in adrenal masses is a crucial topic. This point has been discussed in paragraph 10.1. Imaging, radiomics and biomarkers (page 23).
Round 2
Reviewer 1 Report
The authors have successfully addressed the suggested comments concerning their manuscript. I therefore suggest this manuscript for publication.
Author Response
Thank you for your report. Minor suggested revisions have been applied.
Reviewer 2 Report
No further comments
Seems ok
Author Response

(The authors gave the same response as above.)

Reviewer 3 Report
I think that the authors have adequately addressed the comments. I only suggest to shorten the figure legend, is too long and contains comments not required in a legend. I have no further comments.
Author Response
Thank you for your report. Minor suggested revisions have been applied in figure legend.